# Midline Skull Base Meningiomas: Transcranial and Endonasal Perspectives

**DOI:** 10.3390/cancers14122878

**Published:** 2022-06-10

**Authors:** Ciro Mastantuoni, Luigi Maria Cavallo, Felice Esposito, Elena d’Avella, Oreste de Divitiis, Teresa Somma, Andrea Bocchino, Gianluca Lorenzo Fabozzi, Paolo Cappabianca, Domenico Solari

**Affiliations:** Division of Neurosurgery, Department of Neurosciences and Reproductive and Odontostomatological Sciences, Università degli Studi di Napoli Federico II, 80131 Naples, Italy; mastantuoniciro@gmail.com (C.M.); felice.esposito@unina.it (F.E.); elenadavella@gmail.com (E.d.); oreste.dedivitiis@unina.it (O.d.D.); teresa.somma@unina.it (T.S.); andrea.bocchino@unina.it (A.B.); gianlucalorenzo.fabozzi@unina.it (G.L.F.); paolo.cappabianca@unina.it (P.C.); domenico.solari@unina.it (D.S.)

**Keywords:** meningiomas, skull base, tuberculum sellae, olfactory groove, cavernous sinus, petrous apex, endoscopic endonasal surgery

## Abstract

**Simple Summary:**

Skull base meningiomas have always represented a challenge for neurosurgeons. Despite their histological nature, they may be associated with unfavorable outcomes due to their deep-seated location and the surrounding neurovascular structures. Over time, several corridors have been proposed, each one carrying its own pros and cons. During the last decades, the endoscopic endonasal route has been asserted among the classic routes for a growing number of midline and paramedian lesions. Therefore, the aim of our paper is to present a comprehensive review of the indications and techniques for the management of skull base meningiomas, emphasizing the ambivalent and complementary role of the low and high routes.

**Abstract:**

Skull base meningiomas have always represented a challenge for neurosurgeons. Despite their histological nature, they may be associated with unfavorable outcomes due to their deep-seated location and the surrounding neurovascular structures. The state of the art of skull base meningiomas accounts for both transcranial, or high, and endonasal, or low, routes. A comprehensive review of the pertinent literature was performed to address the surgical strategies and outcomes of skull base meningioma patients treated through a transcranial approach, an endoscopic endonasal approach (EEA), or both. Three databases (PubMed, Ovid Medline, and Ovid Embase) have been searched. The review of the literature provided 328 papers reporting the surgical, oncological, and clinical results of different approaches for the treatment of skull base meningiomas. The most suitable surgical corridors for olfactory groove, tuberculum sellae, clival and petroclival and cavernous sinus meningiomas have been analyzed. The EEA was proven to be associated with a lower extent of resection rates and better clinical outcomes compared with transcranial corridors, offering the possibility of achieving the so-called maximal safe resection.

## 1. Introduction

Skull base meningiomas have always been one of the ultimate gauntlets for neurosurgeons. Despite their histological nature, they may be burdened by an unfavorable surgical outcome due to their deep location and the involvement of vital neurovascular structures. Since the first pioneering series, neurosurgeons rose to the challenge and delivered reports describing the technical nuances and outcomes of the different surgical routes adopted over the years. Each approach had to take into account the two pillars of meningioma surgery: maximizing the extent of resection (EOR) and lowering the recurrence risk while preserving neurological functions. A fortiori, complex tumors require a peer equilibrium between EOR and patients’ outcomes quoad vitam and valitudinem. Therefore, the pursuit of gross total resection (GTR) at all costs can often result in a Pyrrhic victory related to the unbearable peri-operative and long-term complication rates.

During the last decades, neurosurgery embraced the futuristic flourishing of medical sciences and technological exploitation, and thus far, new corridors have been created. Among these, the endoscopic endonasal approach, offering a wide and panoramic view, increased the versatility of the transsphenoidal route and allowed the exposure of different compartments of the skull base, i.e., the suprasellar area (including the third ventricle), olfactory groove, clival and petroclival region, cavernous sinus, and pterygoid fossa, depicting the “low route” as a reliable alternative to the “high route” [1]. 

Considering the above, the aim of our study is to provide an overview of the state of the art of skull base meningioma surgery, emphasizing the ambivalent and complementary role of the low and high routes. 

## 2. Materials and Methods

A comprehensive review of the pertinent literature was performed in order to address surgical strategies and outcomes of skull base meningioma patients treated through transcranial approach, EEA, or both. Three databases (PubMed, https://pubmed.ncbi.nlm.nih.gov/, first access 1 November 2021; Ovid Medline, https://ovidsp.ovid.com, first access 1 December 2021 and Ovid Embase, first access 3 December 2022) have been searched. The keywords used were: “skull base meningiomas” AND/OR “tuberculum sellae meningiomas” AND/OR “olfactory groove meningiomas” AND/OR “anterior cranial fossa meningiomas” AND/OR “clival meningiomas” AND/OR “petroclival meningiomas” AND/OR endoscopic endonasal approach, AND/OR transcranial approach AND/OR endonasal AND/OR craniotomy AND/OR surgical resection AND/OR microsurgery AND/OR transsphenoidal AND/OR sellar surgery AND/OR outcome AND/OR recurrence AND/OR CSF leak. The search was limited to articles published from 1995 to January 2022 in the English language only. All studies were selected based on the following inclusion criteria: (1) series reporting endoscopic endonasal approaches for tuberculum sellae, olfactory groove, petroclival, and cavernous sinus meningiomas; (2) series reporting microsurgical approaches for tuberculum sellae, olfactory groove, petroclival, and cavernous sinus meningiomas; and (3) systematic reviews and meta-analysis of RTCs, cohort and case-control studies of tuberculum sellae, olfactory groove, petroclival, and cavernous sinus meningiomas.

Exclusion criteria were the following: (1) case reports, (2) studies published in languages other than English with no available English translation, and studies with overlapping patient populations; in this latter circumstance, the most recent series was included. 

Papers comparing or describing two or more approaches have been separately considered into the statistics of each.

Two senior authors (D.S., L.M.C.) independently retrieved the studies, which have been included.

### Definition of Surgical Corridors

The routes analyzed have been classified into transcranial and ventral.

The transcranial corridors were further divided into anterior, anterolateral, lateral, and posterolateral.

The anterior approaches encompass unilateral/bilateral subfrontal, orbitofrontal, transbasal, and interhemispheric routes, along with their minimally invasive variants.

The anterolateral corridors include pterional and its variants (namely fronto-lateral, mini-pterional, fronto-orbito-zygomatic, extended pterional, and lateral supraorbital craniotomies) and the pre-temporal transpetrous approaches.

The lateral routes are the anterior petrosectomy through the subtemporal corridor and the posterior petrosectomies, namely translabyrinthine-transcochlear, retrolabyrinthine-transtentorial, and presigmoid petrosal with partial labyrinthectomy.

The postero-lateral routes embed the retrosigmoid approach, along with its extension (namely transtentorial and suprameatal) and the far lateral approach.

The ventral routes include the endoscopic endonasal approaches extended to either lamina cribrosa, planum sphenoidale, or clivus (Table 1).

## 3. Results

The review of the literature provided 328 papers reporting the surgical, oncological, and clinical results of different approaches to the treatment of skull base meningiomas. 

Eighty-seven papers were about olfactory groove meningiomas (OGMs). Fifty-one manuscripts described surgical techniques and outcomes of transcranial routes. The most common microsurgical corridors employed were the anterior (30 papers, 34.5%) and anterolateral (28 papers, 32.4%). Twenty-nine (33.1%) articles addressed the ventral route. Transcranial corridors featured higher rates of GTR (up to 94%) and lower rates of anosmia (9.4%) and cerebrospinal fluid (CSF) leak (up to 6.4%) compared with EEA (up to 70.3%, 95.9%, and 22%, respectively), whilst the transsphenoidal route enables high rates of visual improvement (Figure 1).

Planum sphenoidale/tuberculum sellae meningiomas (TSMs) were addressed in 153 articles. The review suggests their management through microsurgical anterior (39 papers, 25.6%), anterolateral (53 papers, 34.7%), and ventral approaches (61 papers, 39.7%): transcranial and transsphenoidal routes apparently provide similar results in terms of EOR (the GTR rate was 74.5% for EEA and 76% for transcranial), whilst CSF leak (8.6% vs. 2.1%), infection (9.2% vs. 3.1%), and olfaction dysfunction (19.2% vs. 5.1%) rates were higher in the EEA group than in the TC group. Again, higher visual improvement rates were associated with the endoscopic endonasal group (87% vs. 53%) (Figure 2).

Sixty-three articles reported the outcomes and surgical approaches of clival (CMs) and petroclival meningiomas (PCMs). These lesions are treated by transcranial anterolateral (7 papers, 11.4%,), lateral (19 papers, 29.9%), posterolateral routes (28 papers, 45.1%), and ventral corridors (9 papers, 13.6%); transcranial approaches are related to a higher extent of resection (up to 79% vs. 52%) albeit the burden of higher neurologic complication rates compared with EEA (up to 42% vs. 28%), while CSF leak rate was higher for EEA (41.1% vs. 6%) (Figure 3).

Cavernous sinus meningiomas (CSMs) are discussed by 25 papers proposing management based upon transcranial anterolateral (12 papers, 48%), lateral routes (6 articles, 24%), and ventral corridors (7 articles, 28%); the current policy counts on the debulking of the intracavernous aspects of the lesion and the removal of the outer portion, along with bony decompression. Both transcranial and transsphenoidal surgery provide similar and promising results (Figure 4).

The anterior routes have been addressed in 71 studies (19.83), the anterolateral corridors in 122 (34.07%), the postero-lateral approaches in 27 (7.54%), and the lateral approaches in 25 papers (6.98%). The ventral route was analyzed in 113 manuscripts (31.56%) (Figure 5).

The distribution of the approaches per meningioma type is reported in Figure 6.

## 4. Discussion

### 4.1. Olfactory Groove Meningiomas

The choice of the surgical strategy for addressing olfactory groove meningiomas is challenging and pivoting on the attempt to preserve olfaction. Different approaches and several variants have been considered suitable for these lesions: (1) anterior approaches (namely subfrontal) either through unilateral/bilateral frontal craniotomy, transbasal, or interhemispheric; (2) anterolateral access (frontolateral, pterional, fronto-orbito-zygomatic); (3) ventral access through endoscopic endonasal approach (Table 2).

#### 4.1.1. Transcranial Perspective

The anterior route was the most addressed for OGMs in the pertinent literature.

The subfrontal approach through unilateral or bilateral frontal craniotomy is commonly used for large and giant olfactory groove meningiomas: it provides a short surgical corridor and a wide exposure of the lesion with its dural attachment, allowing easier management of the hyperostotic anterior skull base bone and the eventual decompression of optic nerves by means of the optic canals unroofing.In selected cases, based on tumor size and features, it can be extended via orbital rim osteotomy in order to achieve a more basal trajectory, diminishing brain retraction.

The main drawbacks identified by the analysed studies include [2,3,4,5,6,7,8] (i) overexposure and significant retraction of the frontal lobes, with the risks of edema resulting in cognitive and emotional impairment; (ii) late exposure with the related risk of injury to critical neurovascular structures, such as the optic apparatus, ICA, ACA, and ACoA; (iii) the risk of CSF leak and meningitis due to the opening of the frontal sinus.

Minimally invasive variants of this approach have been successfully adopted and proposed by several groups, and amongst them, the eyebrow supraorbital keyhole craniotomy with or without endoscope assistance proved to be safe and effective, with similar surgical and clinical outcomes but lower rates of frontal lobe injury, CSF leak, and anosmia [9,10,11,12,13].

Recently, the keyhole superior interhemispheric approach has been described as a variant of the superior interhemispheric approach used by Wong et al. [14] and Libório Dos Santos et al. [15] to address lesions of the far frontal region (from the cribriform plate to 4–5 cm above the skull base along the anterior falx cerebri).

The transbasal approach is more suitable for tumors extending caudally into the exocranial skull base, as it provides excellent access to both the paranasal sinuses (frontal, ethmoidal, and sphenoidal) and the orbits. Cecchini et al. reported through an extensive review that the early devascularization of the tumor can be achieved through the early coagulation of the anterior ethmoidal arteries [15]. This route enables direct access to the basal dural attachment, although it is associated with prolonged surgical times, risks of long-term cosmetic defects, and higher CSF leak rates [14,15,16,17,18,19]. Furthermore, it shares with other anterior approaches the disadvantage of late visualization of the optic nerves and vascular structures [7].

The anterolateral approaches, namely pterional along with its modifications, such as fronto-lateral, mini-pterional, fronto-orbito-zygomatic, extended pterional, and lateral supraorbital craniotomies, are suitable for handling olfactory groove meningiomas either via the subfrontal or transsylvian route [20,21,22,23,24,25,26].

Compared to other frontal craniotomies, the pterional approach provides several pros, including lower rates of postoperative CSF leak, preservation of the frontal sinus, early CSF release prior to tumor manipulation, early identification and protection of the neurovascular structures, and potential preservation of the olfaction [27]. However, in cases of large and giant meningiomas, and/or when the orbital roofs and the ethmoid bone are widely distant, the pterional corridor presents troublesome access to the contralateral side since it requires more brain retraction for exposing the lesion and handling its dural attachment [15].

In a recent meta-analysis, Feng et al. disclosed that bilateral approaches (bifrontal, transbasal, interhemispheric) carry higher post-operative complication rates (meningitis 1% vs. 0%, mortality 3.2% vs. 0.2%) and similar clinical and oncological outcomes (GTR rate 90.9% vs. 94.6%) compared with unilateral approaches [7], and their results were concurrent with other reports and surgical series [28,29,30].

#### 4.1.2. Endoscopic Endonasal Perspective

The endoscopic endonasal transethmoidal/transcribiform surgery enables direct access to the lesion, allowing early devascularization via coagulation and the cutting of the ethmoidal arteries prior to exposing the lesion. This is more valuable for tumors extending into the paranasal sinuses and/or the orbits and in cases of recurrent lesions that underwent former craniotomies [31,32,33]. EEA relies on the same microsurgical principles used in the transcranial routes, namely internal debulking, capsule mobilization, extracapsular dissection of neurovascular structures, focal coagulation, and capsule removal [34,35,36,37].

Several studies concur in their identification of EEA’s main advantages, a lower trajectory for direct decompression of optic nerves and a straight control of the perforating vessels, thus determining minimal manipulation of the critical neurovascular structures and allowing high rates of visual outcome improvement and fewer morbidities [31,35,38,39,40].

The main drawbacks of EEA are the high risks of postoperative CSF leak [33,41,42,43,44,45] (up to 22%) and anosmia [27,46,47,48]. Nevertheless, the recent refinement of the sealing techniques, mostly relying on autologous materials, aided in improving postoperative CSF leak rates [49,50,51,52,53,54]. As reported by Najafabadi et al. in a recent meta-analysis, the CSF leak rate dropped over time from 22% in 2004 to 16% between 2011 and 2015 and 4% between 2016 and 2020 [45]. Regarding the risk of anosmia, the endoscopic endonasal route is discouraged in patients with preoperative preserved olfaction, as also suggested by recent decision-making algorithms proposed by Ottenhausen et al. for skull base meningioma management [55]. Nevertheless, olfaction saving has been described via using unilateral EEA, thus upholding the contralateral septal mucosa and removing the lesions without visualizing the contralateral olfactory bulb [55].

Moreover, as reported by Koutourousiou et al., tumor size superior to 40 mm, calcification, and absence of cortical vascular cuff may hinder GTR; significant lateral and anterior dural involvement are relative contraindications for EEA [40].

#### 4.1.3. EEA vs. Transcranial

The choice of the proper surgical corridor relies on a peer equilibrium between the extent of resection and the risk of neurological injury. Olfaction preservation is crucial in patients with an intact sense of smell: the actual trend fosters the anterolateral route and its minimally invasive variants for the resection of olfactory groove meningiomas. For resection of olfactory groove meningiomas with normal olfaction extending laterally beyond the lamina papyracea and the ICA. The extended endoscopic endonasal approach is selected primarily for the management of lesions causing loss of smell, without lateral extension and in cases of lamina cribrosa transgression. For tumors harboring significant lateral and anterior extension (to the back wall of the frontal sinus) with lamina cribrosa invasion, either combined, two-step, anterolateral, or bilateral frontal plus EEA approaches are preferred [54].

Several studies disclosed a direct comparison of surgical outcomes and complications between the low and the high route. Purohit et al. showed that EEA was associated with a higher rate of anosmia (95.9%) lower rate of GTR (69.5%) and higher incidences of CSF leak (26.2%) than the transcranial group (37.4%, 91.8%, and 6.2%, respectively) [56].

Shetty et al. demonstrated that transcranial approaches are still preferable over EEA for both the extent of resection and the complication rates: they reported a higher rate of GTR through transcranial corridors along with a lower incidence of CSF leak and meningitis, whilst EEA provided significantly greater rates of visual improvement (80.07% vs. 12.83%). Therefore, they assumed that EEA should be reserved for selected cases harboring preoperative anosmia, in which the primary goal is relieving visual symptoms [57].

### 4.2. Tuberculum Sellae/Planum Sphenoidale Meningiomas

Since Cushing’s first description, countless reports, series, and classifications arose regarding the so-called suprasellar meningioma, namely lesions arising from sellar and juxtasellar regions, planum sphenoidale, and tuberculum sellae [58,59,60,61,62,63]. Recently, Al-Mefty defined specific subsets based on a tumor’s origin, encompassing in the term suprasellar meningiomas those arising from the planum sphenoidale (PS), tuberculum sellae (TS), diaphragma sellae (DS), and anterior clinoid (AC) processes [49,64].

The analysis of the literature suggests the choice among anterior, anterolateral, and ventral corridors as per tumor feature and clinical presentation (Table 3).

#### 4.2.1. Transcranial Perspective

Among the anterior corridors, the bilateral frontal approach provides a wide and direct view of optic nerves, ICA, anterior cerebral arteries, and the anterior communicating artery complex. Therefore, it is preferred in cases of large tumors with the involvement of both optic canals [65]. A further monolateral orbitotomy may be added to gain a broad infero-superior trajectory. As reported by Chokyu, the optic canals may be unroofed, depending on the degree of tumor invasion [66], thus increasing the visual improvement likelihood [67,68].

The main drawbacks that have been identified in the pertinent literature are (i) the high risk of olfactory nerve damage, (ii) brain edema, (iii) cerebral infarction due to frontal lobe manipulation, and superior sagittal sinus ligation. Moreover, CSF leak through the frontal sinus and subsequent meningitis has been considered a major issue of the bifrontal approach for years of cases in the early series. Recently, the use of an autologous fat graft for plugging the frontal sinus has been drastically reducing the CSF leak rate and its consequences [66].

Contrariwise, unilateral frontal and orbitofrontal craniotomies exploit a subfrontal route, avoiding the downsides related to superior sagittal sinus ligation and bilateral frontal lobe retraction [69]. Moreover, the supraorbital keyhole approach, either purely microsurgical or endoscope-assisted, spares frontal sinus opening, further helping in reducing CSF leak rate [70,71,72].

The unilateral subfrontal route achieves the complete exposure of the optic nerve area, including the inferior surface, thus allowing for the preservation of perforators running in the subchiasmatic area: therefore, it provides a more remarkable improvement of visual functions compared with anterolateral corridors [73,74]. Likewise, the unilateral subfrontal pathway, via both supraorbital and frontal approaches, attains a broader view over the planum sphenoidale and tuberculum sellae than a transsylvian corridor.

The anterior interhemispheric approach (AHA) presents the main advantage of enabling a symmetrical view and control of optic apparatus, ICAs and their terminal, and distal collateral branches without extensive retraction [75,76]. Furthermore, its trajectory is directly along the dorsum and diaphragm sellae, thus allowing for the direct oversight of tumor posteroinferior extension. Pertinent literature has reported a GTR rate greater than 90%, with a high rate of visual improvement [77,78,79,80]; nevertheless, AHA is constrained by superior sagittal sinus and bridging veins sacrifice, a high rate of postoperative anosmia, and a lower quality of exposure over the tumor lateral aspects [75].

Anterolateral approaches provide the shortest distance to the sellar region and tumors arising therein. Thus, they have been deemed to be tailored for sellae meningiomas with lateral and posterior extension [81,82,83,84,85].

Anterolateral corridors harbor several advantages: (i) the possibility of harnessing both subfrontal and transsylvian routes; (ii) frontal sinus sparing; (iii) early visualization and access to the optic chiasm and ipsi/contralateral crucial neurovascular structures; (iv) the early opening of arachnoid cisterns and brain relaxation; (v) a low rate of cerebral edema [69].

Contrariwise, some series reported an inadequate control of tumor extending in the ipsilateral optic canal and inferior surface of the chiasm, triggering post-operative visual deterioration, which is a major con [60,67,69,84].

Troude et al. addressed a controversial issue on the side of the approach: their results disclosed better visual outcomes and tumor control at the expense of a higher rate of anosmia via the contralateral approach [86]. Aftahy et al. proposed a decision-making algorithm for transcranial approaches, recommending anterolateral approaches in cases of optic apparatus involvement and multimorbidity, whilst median approaches are preferred in cases of large tumors determining mass effect and brain edema [3].

As reported by Romani et al. the lateral supraorbital approach, as well as the similar frontolateral approach, can provide the same advantages of pterional and fronto-orbitozygomatic craniotomies without exposing the middle fossa [85].

#### 4.2.2. Endoscopic Endonasal Perspective

The rationale of adopting EEA for the management of planum sphenoidale/tuberculum sellae meningiomas relies mostly on the possibility of the better management of inner tumor features: (i) ventral dural attachment on planum sphenoidale, tuberculum sellae, chiasmatic sulcus, limbus sphenoidale, or diaphragma sellae; (ii) growth pattern directed toward the sella turcica, interoptic, and subchiasmatic area (Figure 7).

The “low route” enables early access to the hyperostotic and dural attachment consenting to immediate tumor devascularization, direct exposure of the subchiasmatic area without brain retraction, and a wide decompression of the optic canals without nerve manipulation [64,72,87,88].

The extended transplanum/transtuberculum endoscopic approach allows for the tailoring of the bone removal on antero-posterior and lateral tumor extension [89] and a customized 160° to 180° inferomedial optic canal decompression [90]. Moreover, in a limited number of cases with a high degree of pneumatization superolateral decompression can be achieved [90]. Concerning the removal rates, GTR has been reported in modern series, ranging from 80 to 95% [62,72,87,88].

The major EEA Achille’s heel remains the postoperative CSF leak: the reconstruction of the osteo-dural breach at the suprasellar notch area [91] is further challenging due to its angled shape and irregular borders. Likewise, the opening of suprasellar and lamina terminalis cisterns via the extended approach exposes a greater amount of CSF leak to be sealed, and in case of repair failure, the risk of pneumocephalus and meningitis increases [92]. The introduction of multilayer reconstruction techniques accounting for autologous materials (above all, the Hadad–Bassagasteguy flap), along with clinical strategies aimed at counteracting CSF pulsation [53,93,94], markedly the reduced rate of complications (from 22% to 4%) [44,62,72,87,88,95].

#### 4.2.3. EEA vs. Transcranial

Sellar and suprasellar regions are encircled by vital neurovascular structures, namely carotid arteries, optic nerves, and chiasm, in which ventral bony protuberances and depressions depict the certain landmarks of the ventral approach. de Divitiis et al. suggested the use of EEA for the management of small-to-medium-sized tumors without lateral extension beyond supraclinoid internal carotid arteries and optic nerves, with limited dural attachment, without vascular encasement, and without calcifications [64]. Further reports corroborated their findings, relegating EEA for midline tumors without invasion or with the inferolateral encroachment of optic canals [96,97,98]. Recently EEA, has been further advocated to ameliorate visual functions for tumors in which vascular encasement prevents total removal through both transcranial and transsphenoidal routes [61].

Transcranial approaches are more suitable for tumors extending beyond the polygon entangled by supraclinoid ICAs and optic nerves and chiasm, harboring a firm or calcified consistency and with a superolateral invasion of the optic canals.

Nevertheless, the ventral route warrants several advantages over the transcranial route: such perks emerged alongside the progressive acquisition of expertise in endoscopic endonasal techniques. Firstly, tuberculum sellae meningiomas receive blood supply from dural vessels such as ethmoidal arteries; the ventral route enables early access and the devascularization of tumor feeders, leading to better surgical field visualization and lower blood loss [99,100]. Secondly, EEA allows for a wider working window with fewer blind spots from ICA to ICA [97]. Several surgical series, systematic reviews, and meta-analyses disclosed that EEA and transcranial routes provide similar EOR, neurological, and endocrinological complication rates, whilst EEA is associated with higher visual function improvement and, as a drawback, higher CSF leak rates [61,87,96,97]. Graffeo et al. reported lower recurrence rates for transcranial approaches [101], whereas Muskens et al. described higher rates of vascular injury for EEA [102].

### 4.3. Clival and Petroclival Meningiomas

Clival and petroclival meningiomas represent among the most complex and thrilling surgical challenges for neurosurgeons. True “clival meningiomas” originate from a broad attachment to the median upper two-thirds of the clivus, and petroclival meningiomas are described as tumors in which the dural attachment is located at the upper two-thirds of the clivus, lateral to the midline, at the level of the petroclival junction, and medial to the trigeminal nerve [103]. Over the years, several classifications arose to warrant different surgical routes aimed at maximizing surgical outcomes while restraining complications [104,105,106,107]. The main approaches described in the pertinent literature rely on the anterolateral, lateral, and posterolateral corridors (Table 4). Early devascularization, as the primary goal for the intraoperative management of skull base meningiomas, is extremely difficult to achieve through both transcranial and endoscopic endonasal routes for PCM. Therefore, the best microsurgical approach should provide a trajectory parallel to the major axis of the tumor.

The peculiar anatomy of the surrounding structures, often engulfed by the tumor, addresses the surgical goal toward a more conservative “function sparing” resection followed by radiotherapy [108,109,110,111,112,113,114]. In this vein, most historical and invasive strategies have been sidelined in favor of less demolitive approaches [103,115,116,117,118,119] and, recently, the improvement in surgical expertise and confidence with the use of endoscope via the transsphenoidal route, which allowed us to enclose clival and some petroclival meningiomas into the possibilities of this approach.

#### 4.3.1. Transcranial Perspective

The retrosigmoid approach (RSA), as the epitome of posterolateral approaches, is the most suitable for petroclival meningiomas projecting toward the cerebellopontine angle and invading the internal acoustic meatus, with no significant extension into the middle cranial fossa. This route allows early brainstem decompression, which is recognized as a crucial step for extra-axial infratentorial pathologies. Moreover, RSA can be extended superiorly by means of the transtentorial variant [120,121,122], anteriorly up to the Meckel’s cave via intradural suprameatal bone removal [121,123], and inferiorly by drilling the lateral portion of the occipital condyle [120]. Purely clival meningiomas and lesions spreading further below the jugular meatus can be approached via a far lateral approach [114].

The major advantage of the retrosigmoid approach is represented by the avoidance of extensive petrous bone resection and a reduced risk for VII and VIII cranial nerves palsy due to their direct visualization.

Transpetrosal approaches enable a wide skull base exposition and visualization of clival and petroclival regions: the extent of the petrosectomy depends on the invasion of the cavernous sinus, Meckel’s cave, the internal auditory canal, the supratentorial extension of tumor, and pre-operative functional hearing [124,125].

Although hearing should be preserved whenever possible, a total petrosectomy may be warranted even in patients with intact hearing in cases of midclival tumors [126,127].

The anterior petrosectomy designed by Kawase et al. [128] provides good visualization and discrete surgical freedom on the middle fossa, Meckel’s cave, the upper third of clivus, and ventral portion of the brainstem. Nevertheless, some authors have expressed their concerns in regard to the size of the window in the petrous bone, which is considered inadequate for posterior fossa exposure [128,129,130].

Therefore, several variants arose recently for customizing the approach upon the lesion and improving cavernous sinus exposure via also exploiting anterolateral surgical corridors, as seen in the intradural petrosectomy described by Steiger et al. [131,132,133] and its variants [134,135].

Posterior petrosectomy enables a broad view of the middle clivus and early access to the tumor feeding vessels, granting an extended view over the posterior fossa compared with the anterior petrosal approach [136,137,138,139].

The posterior petrosal translabyrinthine-transcochlear approach offers the largest corridor at the price of total hearing loss and a certain degree of facial nerve palsy [119,136,137,138]. Moreover, the sigmoid sinus is largely exposed up to the jugular bulb in order to allow its mobilization: thus, a dominant or single sigmoid sinus on the side of the operation, the disconnection of the transverse sinus from the torcular, and an abnormal vein of Labbe anatomy are considered relative contraindications for the approach.

Lesser invasive petrosectomies, namely the retrolabyrinthine-transtentorial and the standard presigmoid petrosal approach with partial labyrinthectomy, have been proposed, allowing good rates of hearing preservation but drastically reducing the surgical exposure [139,140,141,142,143].

A combined transpetrosal approach can be exploited for lesions growing medially to the internal auditory canal and spanning both middle and posterior cranial fossae [116,144].

Anterolateral approaches for petroclival meningiomas rely on the combination of pterional craniotomy with the Dolenc technique [145,146,147] for exposing cavernous sinus, as described by Couldwell et al. [148]: this route involves extradural anterior clinoidectomy and the peeling of the temporal fossa and gives access to the Meckel’s cave and the cavernous sinus [149]. Furthermore, by removing the tumor inside the Meckel’s cave, a further corridor toward the posterior fossa is unlocked [150].

In 2020, Zhao et al. proposed a novel classification for petroclival meningiomas reporting a large surgical series, recommending the most suited surgical approaches for each subset [105]. Lesions encroaching the cavernous sinus may be managed through the anterior transpetrosal approach, pretemporal transcavernous approach and the pretemporal transcavenrous anterior transpetrosal approach, whilst the extended pterional transtentorial approach can be proper for tumors arising from the lateral wall of the cavernous sinus extending both inside and outside it [105].

#### 4.3.2. Endoscopic Endonasal Perspective

EEA can be a valid surgical option for clival meningiomas and also for those that present quite a large size [114,151], as it provides direct access to and visualization of the tumor dural attachment and the obliteration of its blood supply without brain retraction and extensive bone removal. Contrariwise, paramedian lesions such as petroclival meningiomas are considered the most complex and technically demanding among the endoscopic ventral corridors [152,153,154]. The “anterior transpetrosal approach” is performed as previously described via combining transpterygoid and transclival approaches [155,156,157]. The petrous apex is then drilled inferiorly to the horizontal ICA in a caudal-to-rostral and medial-to-lateral direction, with the vidian canal representing the superior limit of this portion of the dissection [158,159,160,161] in order to ensure wide ventral exposure of the petroclival area through the carotid-basilar corridor [154,162]. Furthermore, a contralateral transmaxillary (CTM) trajectory may be added to the aforementioned exposure, enlarging the surgical view and administration of the petrous segment of the internal carotid artery [163,164,165]. Pituitary transposition can be completed to access the interpeduncular cistern and perform a posterior clinoidectomy by removing the dorsum sellae [166,167,168,169]. Recently, a transcavernous technique has been proposed to overcome pituitary dysfunctions but is technically demanding and adds further blood loss [169,170].

EEA is usually adopted for managing petroclival meningiomas spreading in a medial-to-lateral direction, remaining medial to the cranial nerves’ foramina and the internal carotid arteries, exploiting the route between the basilar artery and ICA [154]. It allows direct access to the medial tumor attachment with an early devascularization by coagulating the clival and petroclival feeders, namely meningeal arteries and lateral and medial clival arteries. EEA grants a wide and direct visualization of the abducens nerve, vertebrobasilar system, brainstem, and lower cranial nerves, easing their dissection from the tumor [171].

The use of EEA is integrated into a modern paradigm aimed at balancing the surgical results with clinical outcomes, therefore, its main goal is hardly ever a complete resection but an effective brainstem decompression with minimal neurovascular manipulation [114,172,173].

The endoscopic endonasal approach’s major con is a high CSF leak rate, ranging up to 41.2%: this issue is even harder to manage in the petroclival area due to the peculiar conformation and the size of osteodural removal [114]. Recently, a septal rhinopharyngeal flap has been proposed, using septal, rhinopharyngeal roof, and posterior wall mucosa and allowing for the coverage of the tuberculum/sellar region, mid-clivus, and lower clivus, with promising results [174].

#### 4.3.3. EEA vs. Transcranial

The overall GTR rates of PCMs reported in the literature were ranging from 20% to 79% [103,105,112,116,143,175,176,177], being the lowest EOR addressed by endoscopic endonasal route [114,172]. EEA proved to be useful for addressing lesions located medially to skull base foramina, whilst transcranial routes are suitable for tumors arising lateral to them. Moreover, infratentorial tumor removal cannot benefit from a cranio-caudal trajectory hindered by osseous, vascular, and neural structures, so the endoscopic endonasal treatment for posterior fossa meningiomas provides the advantage to override this issue through a direct route toward petroclival tumors not extending beyond the medial border of the hypoglossal canal, and jugular foramen and internal carotid artery.

Another crucial issue is the management of the basilar artery and VI cranial nerve [114,178]: EEA allows early identification of these structures, whilst conventional craniotomies expose them in the latest stage of the tumor debulking. Van Gompel et al. compared the petrosectomy extension between EEA and transcranial approaches, retrieving data to support that transcranial anterior petrosectomy furnishes a larger area of exposure of the upper petroclival area, while endoscopic ventral petrosectomy delivers a better exposure of the medial and inferior portion of the petroclival junction overlying the inferior petrosal sinus. Therefore, they redefined both skull base osteotomies as “superior anterior petrosectomy” and “inferior anterior petrosectomy” [165].

Considering the limits and assets of both ventral and transcranial corridors, a combined approach is being successfully used to manage complex lesions whose extension and location preclude adequate debulking via a single route [153].

In the pertinent literature, the extent of petroclival meningiomas resection presents a broad range from 20 up to 79%, as reported by Koutourousiou et al. [114], being the higher rates of GTR associated with more invasive approaches and higher rates of complications and vice versa [103,105,108,118,177,179,180,181]. In this scenario, EEA provides a lower resection rate, but it determines a higher rate of early clinical recovery and less complications than transcranial approaches [114].

### 4.4. Cavernous Sinus Meningiomas

“*A meningioma involving the cavernous sinus is an ordinary event in an extraordinary site that is inhabited by closely packed vital nerves and vessels*”.[182]

Purely cavernous sinus meningiomas are extremely rare diseases accounting for less than 1% of all intracranial tumors, nevertheless, they still represent a deal-breaker for neurosurgeons being burdened by undue morbidity and mortality rates. Since the first anatomical reports and surgical attempts performed by pioneers like Dolenc, Hakuba, and Parkinson, the treatment strategy markedly evolved.

The current treatment strategy relies on the tumor extracavernous resection and intracavernous debulking and decompression followed by adjuvant radiosurgery, as firstly suggested by Couldwell et al. [110,150,183,184,185,186,187,188,189,190,191,192,193,194,195,196,197,198,199,200,201,202,203,204,205,206,207,208,209,210,211,212]. Surgical decompression determines the improvement of preexisting neurological symptoms against a negligible complication occurrence [188,190,206,213,214,215]. *A fortiori*, cavernous sinus meningiomas have an indolent growth rate, thus not requiring aggressive management for preventing recurrences [216,217,218]. Patients with asymptomatic or minimally symptomatic CSMs without any visible growth may be candidates for the watchful waiting option [185,210,216,219].

The surgical routes claimed so far to access the cavernous sinus and the parasellar area are (i) lateral, (ii) anterolateral, and (iii) ventral corridors (Table 5).

#### 4.4.1. Transcranial Perspective

The anterolateral route by Dolenc counts on the removal of the temporal floor up to the carotid canal, orbital roof, and anterior clinoidectomy for achieving a good visualization of the sellar and parasellar region and good control of the carotid artery and optic nerve with minimal brain retraction [220,221,222,223,224,225,226,227].

Therefore, the modern anterolateral routes involve the extended pterional and fronto-temporo-orbito-zygomatic approaches, supplemented by the anterior clinoidectomy, the unroofing of the optic nerve, and the extradural exposure of the cavernous sinus.

FTOZ has been shown to increase the surgical freedom and the projection angle while decreasing the amount of brain retraction, allowing an adequate exposure for lesions involving the middle fossa and adjacent areas [228,229,230,231]. An extended pterional approach is suitable for managing spheno-cavernous meningiomas and sphenoclinoidocavernous meningiomas as well as small tumors confined in the lower part of the cavernous sinus. In the latter case, a further osteotomy of the zygomatic arch may be adequate [190,232]. Often, a combined intra/extradural approach may be required to deal with tumors expanding in the lateral wall of the cavernous sinus and infratemporal fossa through cranial nerves foramina, and to decompress cranial nerves since they pierced the roof of the cavernous sinus [210,211,212,213,214,215,216,217,218,219,220,221,222,223,224,225,226,227,228,229,230,231,232,233].

Chen et al. described the pretemporal trans-cavernous, trans-Meckel’s, trans-tentorial trans-petrosal approach for addressing lesions extending from the cavernous sinus, Meckel’s cave, and optic foramen to the posterior fossa. This corridor tops up the incomplete view provided by the transsylvian avenue through a complete visualization of the middle fossa and pre-cavernous course of the cranial nerves, also freeing them from tentorial restrictions [234].

Lateral approaches account for anterior, posterior, and combined petrosectomy. The removal of the petrous bone is mostly required for petroclival tumors extending into the cavernous sinus. This osteotomy enables pregasserian and retrogasserian corridors, allowing for the exposure and decompression of the CNs from the brainstem to the CS [105,219].

Morisako et al. popularized the minimal anterior and posterior combined transpetrosal approach, involving a temporooccipito-suboccipital craniotomy and exposure of the posterolateral part of the CS via a presigmoidal anterior and posterior petrous bone removal [235].

A 2010 meta-analysis compared the efficacy of gross total resection, subtotal resection, and SRS alone in 2065 patients with CSMs and found that SRS resulted in a significantly reduced recurrence rate as well as a reduced rate of postoperative nerve deficits. No statistically significant differences between the gross total and subtotal resection were founded [236], and the tumor control rate ranged from 81% to 94.1% over a period of 2–8.3 years. The complication rates for aggressive transcranial surgery were high, reaching up to 59.6%, and included cranial nerve and vascular injuries, ischemia, pituitary dysfunctions, and death. Likewise, it hardly ever led to the improvement of preexisting symptoms [198,236,237,238,239].

One reasonable explanation is that tumor dissection from the neurovascular structures may disrupt their delicate blood supply [168,240]. Considering the above, the paradigm has shifted toward the extracavernous lesion removal and debulking and decompression of the intra-cavernous component, also simplifying the surgical approaches and reducing bone removal [188,190,206,213,214,215].

#### 4.4.2. Endoscopic Endonasal Perspective

The use of EEA for the management of cavernous sinus meningiomas is backed by pioneeristics’ anatomical studies performed by Cavallo et al. [241], Alfieri et al. [242,243], Rhoton et al. [244], Frank et al. [245] and Cappabianca et al. [246], which defined new safe triangles for accessing cavernous sinus, in both its medial and lateral compartments via a ventral corridor. Three endoscopic endonasal trajectories have been advocated to enable a tailored exposure, namely the extended endoscopic endonasal transsphenoidal, endonasal transethmoidal/transsphenoidal (far lateral), and contralateral endoscopic endonasal transsphenoidal approaches [244,247]. The ipsilateral and far lateral routes unveil the surgical triangles lying laterally to the ICA, whilst the contralateral approach is suitable for exploring the area medial to the ICA [241,242,243].

EEA provided some major advantages: (i) the direct visualization over the bony prominences of nerves and vessels contributes to obviating the risk for neurovascular injury; (ii) the medial wall of the cavernous sinus is made up of only a thin and inconstant dural layer, thus depicting a sliding door that allows the surgeon to follow and debulk the tumor within the cavernous sinus [248,249]; (iii) early relief of the pituitary gland with a concurrent improvement of its function [213]; (iv) a direct infero-medial decompression of the optic canal [187]; (v) the approaches to the cavernous sinus and to the sella do not expose subarachnoid spaces; therefore, the risk of CSF leak is extremely low (Figure 8).

The current scenario for using the ventral route for cavernous sinus treatment foresees a conservative transsphenoidal decompression through the removal of the bone overlying sellar and cavernous sinus regions, dural opening, and debulking of the tumor. Furthermore, a 180° medial optic nerve decompression can be performed if needed [187,213,215,250,251].

Hence, EEA, integrated with adjuvant treatment strategies, guarantees optimal results in terms of tumor control and neurological symptoms relief ranging from 42 to 77% [187,213,215,250,251]. Besides, these results were consistent with the outcomes of conservative transcranial management [188,190].

#### 4.4.3. EEA vs. Transcranial

Cavernous sinus meningiomas represent an extremely complex pathology, whose erratic behavior hinders a univocal management strategy.

The current attitude toward a conservative surgical strategy should be considered to address the most suitable route for each tumor: transcranial approaches might be preferred for lesions extending beyond the cavernous sinus into the middle cranial fossa. The extracavernous portion of the meningiomas shall be completely removed, and the intracavernous lesion debulked. Moreover, the optic canal can be unroofed, thus allowing an early relief of the involved cranial nerves. Contrariwise, EEA is the most fitting approach for medial decompression, in cases of pituitary dysfunction or inferomedial invasion of the optic canal.

Both transcranial and transsphenoidal surgery for decompressive purposes provide similar results, with high rates of tumor control (ranging from 90 to 100%) and neurological improvement (ranging from 48% to 66%).

### 4.5. Management Evolution

Skull base meningiomas are polyhedral benign diseases arising in ultimately complex anatomical areas. The main goal of the treatment is to attain long-term tumor control and symptom relief without the burden of severe complications. In order to achieve this goal, neurosurgeons must be versatile, receptive to innovations, and aware of their forerunners’ teachings.

Throughout the years, many excellent contributors paved the way for refining the surgical techniques that ruled out modern skull base surgery, embedding deep anatomical insights, and technological innovations.

During this path, two currents arose: the first was the transcranial route that numbers among its forefathers, Sir Victor Horsley, Harvey Cushing, Walter Dandy with George Heuer, Charles Frazier, Herbert Olivecrona, Francesco Castellano, Beniamino Guidetti, John Jane, and Collin McCarty. Alongside this, a hazardous corridor was gaining land: following Durante’s lead, pioneers, such as Harvey Cushing, Schloffer, von Eiselsberg, Norman Dott, Gerard Guiot, and Jules Hardy, employed exocranial routes to access the skull base from below [252,253,254].

Subsequently, the introduction of the microscope first and the endoscope later, significantly pushed neurosurgery beyond its previous limits: the microsurgical era embraces both the transcranial, with the tremendous contributions of Parkinson, Dolenc, Sekhar, Al-Mefty, Bernard George, Yasargil, and others, and the transsphenoidal routes started by Hardy [143,253,254,255,256].

In 1987, Weiss, fostered the exocranial approaches by developing the extended transsphenoidal route and creating a new paradigm for unlocking the suprasellar supradiaphragmatic space via a ventral route [52,56,257,258].

The deployment of the endoscope in 1996, based upon the fruitful cooperation between Hae-Dong Jho, a neurosurgeon, and the otorhinolaryngologist Ricardo Carrau, further bolstered the transsphenoidal route, and thus, the endoscopic era began [259,260]. Many surgeons deployed and disclosed this route sharing their surgical expertise worldwide; among them, Cappabianca headed the endoscopic revolution from Naples, harnessing the endoscopic endonasal route for treatment of suprasellar meningiomas [1,64,261].

His contribution, sublimated by the publication of “Meningiomas of the skull base”, followed the Francesco Castellano neurosurgical pillar “Meningiomas of the posterior fossa”, allowing the Neapolitan School to be at the forefront in the history of meningioma surgery.

Nowadays, the ventral route has generally been accepted as a viable operative corridor for the management of some cranial base lesions.

The endoscopic endonasal route provides the shortest and most direct access to the lesions gaining maximal exposure, minimal risk for brain retraction, and neurovascular complications. Furthermore, the ventral corridor with the extended endonasal variants enables an early devascularization of the tumor via straight access to its dural attachment, unlocking most of the midline and paramedian skull base. EEA is particularly worthwhile in cases of recurrent lesions from former craniotomies, where it represents a naive route, providing the possibility to bypass the adherences and to avoid further brain manipulations [31,32,33].

Ultimately, the major EEA asset is that it translates deep-seated skull base lesions into superficial convexity-like tumors, making the surgical trajectory “coaxial” with the direction of tumor extension [88,262,263].

The main hindrance of EEA remains the higher rate of postoperative CSF leakage.

Over the last years, different reconstruction strategies contributed to significantly decreasing the overall leakage rate to nearly 5% [263,264,265,266,267,268,269]. The current policy involves the use of autologous material, namely fat pads and nasoseptal or rhinopharyngeal flaps and/or similar tissues. Backing upon these concepts, our group developed the 3F technique, with each F addressing a critical moment of the reconstruction strategy; the first F stands for autologous fat, the second for the naso-septal *flap*, and the third for our idea of *flash*, or early patient mobilization after surgery. This strategy moved further close to zero the likelihood of postoperative CSF leakage [53].

The pertinent literature reports a lower rate of complete resection using EEA as compared with the transcranial route for all skull base meningiomas, except for tuberculum sellae/planum sphenoidale lesions, in which the resection rate is similar via both the transsphenoidal and the transcranial approaches. However, the direct access to critical neurovascular structures and their early decompression allows EEA to achieve equal or better results in terms of neurological and endocrinological improvement and complications. Above all, the ventral route for olfactory groove and suprasellar meningiomas provide better visual outcomes than the transcranial route. Conversely, the transcribriform approach for olfactory groove meningiomas is discouraged. These differences may be accrued by the larger exposure obtained via transcranial approaches, albeit that such board tumor visualization and aggression are upheld by extensive bony removal, undue brain retraction, and the violation and sacrifice of nobles and not pathological structures, namely cochlea, Sylvian fissure, superior petrosal sinus, and Dandy’s vein. Thus, the better extent of resection and tumor control achieved by transcranial corridors may be at the expense of higher rates of long- and short-term complications, longer hospital stays, and delayed recovery.

Therefore, the modern surgical attitude toward skull base meningiomas is going through a paradigm shift focused on the concept of “maximal safe resection”. The evolution of adjuvant therapies relocated surgeons’ awareness of clinical outcomes over the extent of resection. Indeed, a “maximal safe resection”, along with proper secondary treatment, proved to grant optimal long-term tumor control with lower morbidity rates. *A fortiori*, adjuvant therapies, such as radiotherapy or radiosurgery, play a pivotal role in cases of grade II or III meningiomas, according to the WHO classification [202,251,270,271,272,273]. The endoscopic endonasal approach fits this new demeanor, as it provides optimal clinical results in terms of high neurological improvement and low morbidities rates, even more for challenging lesions like petroclival and cavernous sinus meningiomas.

Among the limitations of this study, we perceived the lack of an overview of modern surgical routes, such as the trans-orbital and multi-portal approaches. Although we strongly believe in their primary role in the modern paradigm of skull base meningioma management, the current literature provided only a few studies with often overlapping cohorts. Therefore, we preferred not to include their preliminary results in our paper.

## 5. Conclusions

The treatment of skull base meningiomas relies on multiple approaches belonging to both transcranial and transsphenoidal corridors. Endoscopic endonasal routes enable a wide visualization and direct access to the lesions without brain manipulation; they feature a lower extent of resection rates and better clinical outcomes, as compared with transcranial corridors, but it offers the possibility of achieving, in the vast majority of cases, the so-called maximal safe resection.

We entrust that the concept of adoptive EEA in multidisciplinary management, sacrificing EOR and favoring clinical outcomes, might be considered the essence of neurosurgery evolving toward the quest for smart policies instead of virtuoso performers.

## Figures and Tables

**Figure 1 cancers-14-02878-f001:**
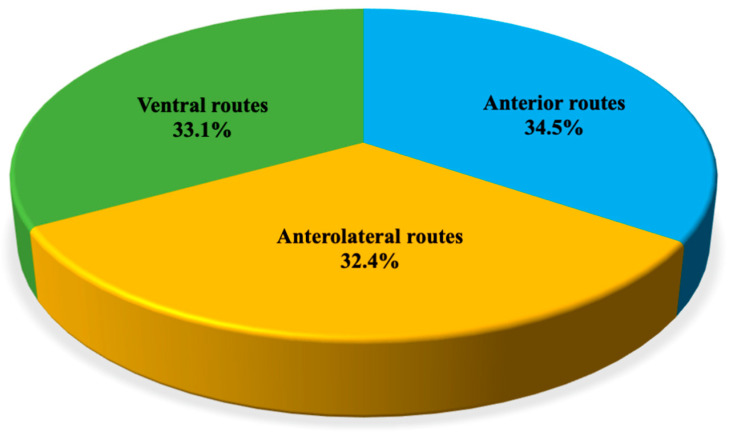
Distribution of papers addressing olfactory groove meningiomas (OGMs) per surgical corridor.

**Figure 2 cancers-14-02878-f002:**
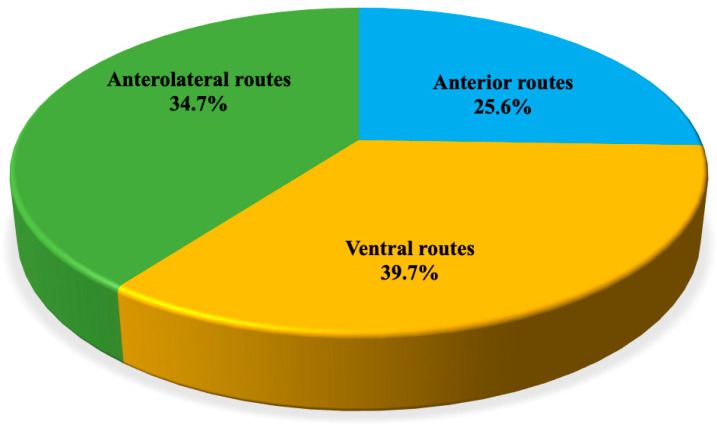
Distribution of papers addressing tuberculum sellae meningiomas (TSMs) divided per surgical corridor.

**Figure 3 cancers-14-02878-f003:**
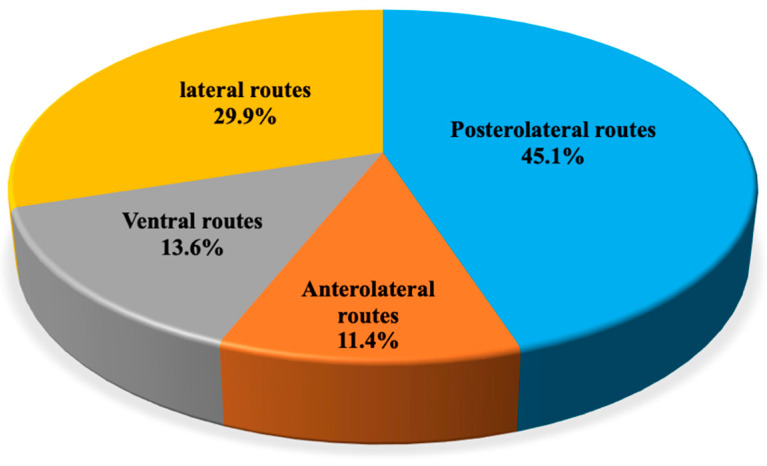
Distribution of papers addressing petroclival meningiomas divided per surgical corridor.

**Figure 4 cancers-14-02878-f004:**
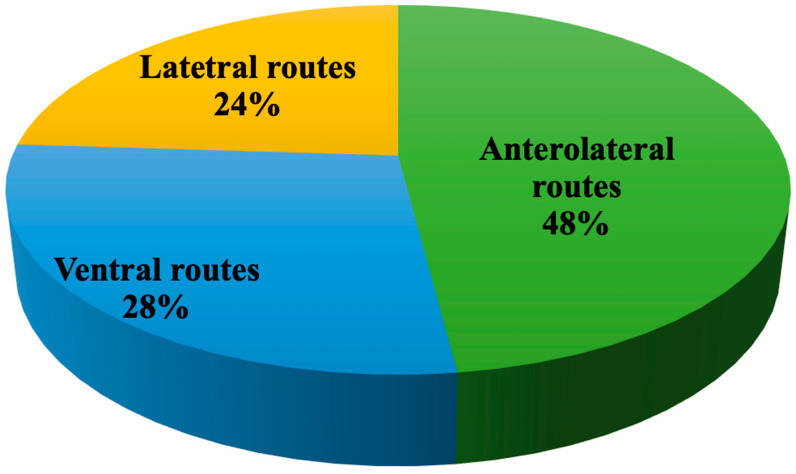
Distribution of papers addressing cavernous sinus meningiomas divided per surgical corridor.

**Figure 5 cancers-14-02878-f005:**
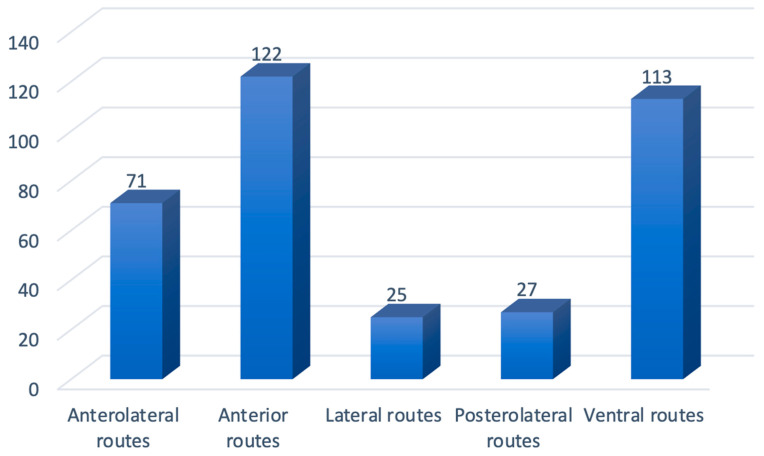
Overall distribution of papers per approach.

**Figure 6 cancers-14-02878-f006:**
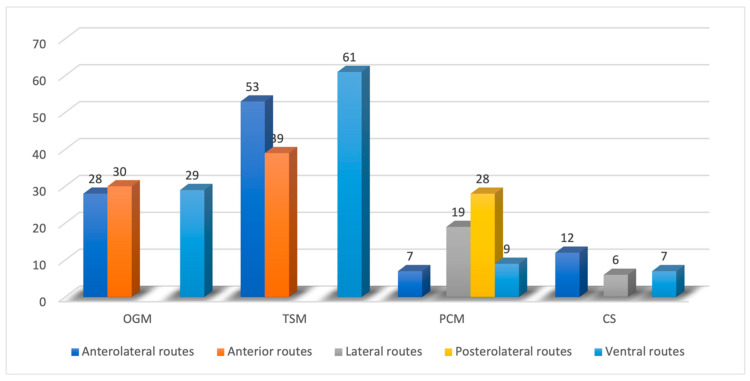
Distribution of employed routes per type of tumor.

**Figure 7 cancers-14-02878-f007:**
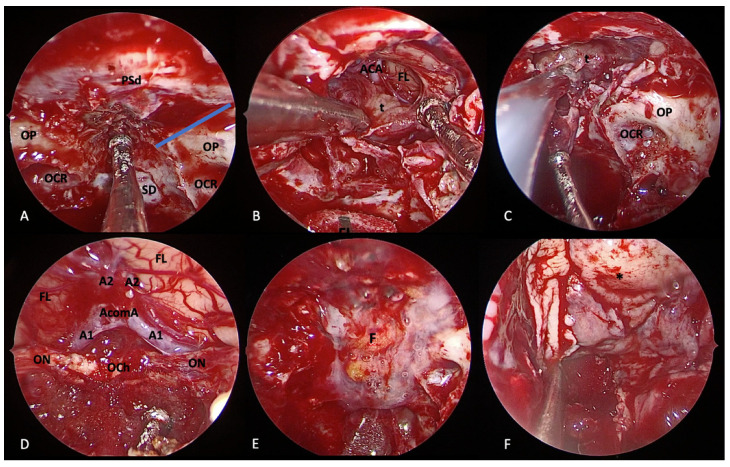
Extended endoscopic endonasal transplanum/transtuberculum approach for removal of tuberculum sellae meningioma. (**A**) Margins of bone removal: laterally, the optic nerves and the optocarotid recess; inferiorly, the sellar floor; anteriorly, the bone removal is tailored on tumor extension. Extracapsular dissection (**B**) and internal debulking (**C**) of the tumor. Surgical field view after tumor removal (**D**). Closure is performed using fat (**E**), nasoseptal flap (**F**), and fibrin glue. Tumor (t); dura of the planum sphenoidale (PSd); optic protuberance (OP); optocarotid recess (OCR), sellar dura (SD); frontal lobe (FL); anterior cerebral artery (ACA); optic nerve (ON); optic chiasm (OCh); fat (F); nasoseptal flap (*).

**Figure 8 cancers-14-02878-f008:**
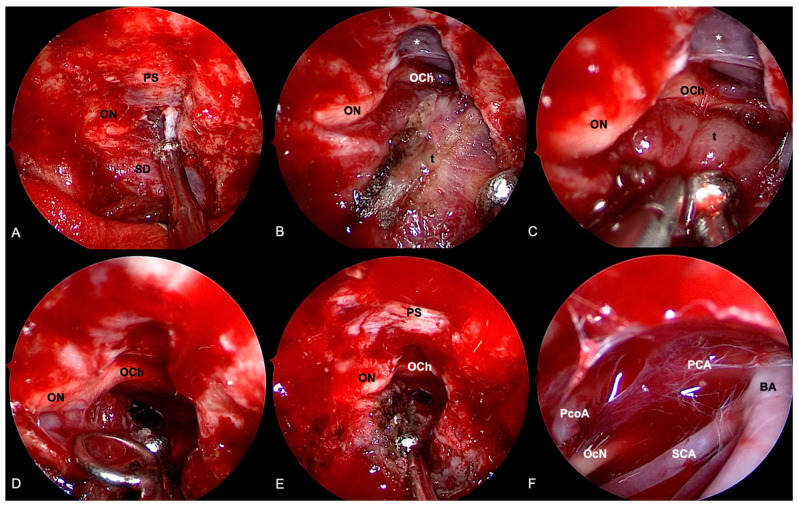
Extended endoscopic endonasal approach for removal of right cavernous sinus meningioma. (**A**) Bone removal encompasses the opening of the mesial aspect of the right optic foramen. (**B**) Sellar dura and planum sphenoidale are exposed. The tumor is coagulated, and the removal proceeds by means of intra and extracapsular dissection. A cleavage plane is found between the lesion and the optic chiasm (**C**) and nerve (**D**). (**E**) After optic nerve and chiasm decompression, the tumor is not followed inside the cavernous sinus. (**F**) Exploration of the interpeduncular cistern through a subchiasmatic window: Basilar artery (BA) and right posterior cerebral artery (PCA), posterior communicating artery (PcoA), and oculomotor nerve (OcN) are identified. Tumor (t); sellar dura (SD); planum sphenoidale (PS); optic nerve (ON); optic chiasm (OCh); oculomotor nerve (OcN); posterior communicating artery (PcoA); posterior cerebral artery (PCA); basilar artery (BA); superior cerebellar artery (SCA); lamina terminalis cistern (*).

**Table 1 cancers-14-02878-t001:** Definition of the surgical corridors.

**Anterior Routes**
unilateral/bilateral subfrontal approach orbitofrontal approach transbasal approach interhemispheric approach
**Anterolateral Routes**
Pterional ○ fronto-lateral ○ mini-pterional ○ fronto-temporo-orbito-zygomatic ○ fronto-orbito-zygomatic ○ extended pterional ○ lateral supraorbital craniotomies ○ pre-temporal transpetrous approaches
**Lateral Routes**
anterior petrosectomy posterior petrosectomies ○ translabyrinthine-transcochlear ○ retrolabyrinthine-transtentorial ○ presigmoid petrosal with partial labyrinthectomy ○ Combined petrosectomies
**Posterolateral Routes**
Retrosigmoid approach ○ intradural suprameatal approach ○ transtentorial approach Far-lateral approach
**Ventral Approach**
Endonasal trans-planum-transtuberculum ap-proach Endonasal transethmoidal transcribiform approach Extended endoscopic endo-nasal transsphenoidal endonasal transethmoid-al/transsphenoidal (far lateral) contralateral endoscopic endonasal transsphenoidal approach Endonasal transclival transpterygoid approach

**Table 2 cancers-14-02878-t002:** Pros and cons for each corridor for the management of olfactory groove meningiomas (OGM).

**Corridor**	**Pros**	**Cons**
**Anterior Route**		
Bilateral subfrontal approach	short surgical corridor wide exposure management of the hyperostotic anterior skull base bone optic canals unroofing	overexposure and significant retraction of the frontal lobes late exposure and risk of injury of optic apparatus, ICA, ACA, and ACoA risk of CSF leak and meningitis due to the opening of frontal sinus superior sagittal sinus ligation bilateral frontal lobe retraction
Unilateral subfrontal approach	No superior sagittal sinus ligation No bilateral frontal lobe retraction No frontal sinus opening short surgical corridor wide exposure management of the hyperostotic anterior skull base bone optic canals unroofing	late exposure and risk of injury of critical optic apparatus, ICA, ACA, and ACoA
Transbasal approach	early devascularization of the tumor (early coagulation of the anterior ethmoidal arteries) excellent access to paranasal sinuses and the orbits	late exposure and risk of injury of critical optic apparatus, ICA, ACA, and ACoA prolonged surgical times risks of long-term cosmetic defects High CSF leak rate
**Anterolateral Route**		
Pterional approach and its variants	lower rates of postoperative CSF leak preservation of the frontal sinus early CSF release early identification and protection of the neurovascular structures preservation of the olfaction	difficult access to the contralateral side in cases of large/giant tumors
**Ventral Route**		
Endonasal transethmoidal transcribiform approach	Early tumor devascularization Direct access to paranasal sinuses and/or the orbits direct decompression of optic nerves straight control of the perforating vessels	high risk of postoperative CSF leak Anosmia

CSF: cerebrospinal fluid; ICA: internal carotid artery; ACA: anterior cerebral artery; ACoA: anterior communicating artery.

**Table 3 cancers-14-02878-t003:** Pros and cons for each corridor for the management of tuberculum sellae meningiomas (TSM).

Corridor	Pros	Cons
**Anterior Route**		
Bilateral subfrontal approach	short surgical corridor wide exposure management of the hyperostotic anterior skull base bone optic canals unroofing Better visualization in cases of large/giant tumors Exposure of optic nerves inferior surface and preservation of perforators	overexposure and significant retraction of the frontal lobes risk of CSF leak and meningitis due to the opening of frontal sinus superior sagittal sinus ligation bilateral frontal lobe retraction
Unilateral subfrontal approach	No superior sagittal sinus ligation No bilateral frontal lobe retraction No frontal sinus opening short surgical corridor wide exposure management of the hyperostotic anterior skull base bone optic canals unroofing Exposure of optic nerves in-ferior surface and preserva-tion of perforators	
Anterior interhemispheric approach	symmetrical view and control of optic apparatus, ICAs and their branches No extensive retraction direct view of tumor postero-inferior extension	superior sagittal sinus and bridging veins sacrifice high rate of postoperative anosmia inadequate exposure of the tumor lateral margins
**Anterolateral Route**		
Pterional approach and its variants	possibility to use both subfrontal and transsylvian routes lower rates of postoperative CSF leak preservation of the frontal sinus early CSF release early identification and protection of the neurovascular structures preservation of the olfaction	inadequate control of tumor extending in the ipsilateral optic canal and inferior surface of the chiasm
**Ventral Route**		
Endonasal transplanum-transtuberculum approach	Early tumor devascularization direct exposure of the subchiasmatic area without brain retraction direct decompression of optic nerves straight control of the perforating vessels	high risk of postoperative CSF leak [33,41,42,43,44,45]

CSF: cerebrospinal fluid; ICA: internal carotid artery.

**Table 4 cancers-14-02878-t004:** Pros and cons for each corridor for the management of petroclival meningiomas (PCM).

Corridor	Pros	Cons
**Lateral Route**		
Anterior petrosectomy	Acceptable surgical freedom on MCF, MC, upper third of the clivus and ventral brainstem	Inadequate bony window for accessing PCF
Posterior petrosectomy	broad view of the middle clivus early access to the tumor feeding vessels extended view over the posterior fossa	The greater the exposure the greater the risk of hearing loss and facial damage Wide exposure and manipulation of the sigmoid sinus
Combined transpetrosal	exploited for lesions spanning both middle and posterior cranial fossae	-
**Posterolateral Route**		
Retrosigmoid approach and its variant	early brainstem decompression middle fossa extention (transtentorial variant) MC extention (suprameatal drilling) early CSF release avoidance of extensive petrous bone resection reduced risk for VII and VIII cranial nerves palsy	inadequate control of tumor extending into the CS
**Anterolateral Route**		
Pretemporal trancavernous anterior transpetrosal approach Extended pterional transtentorial approach	Better control of tumor inside CS and MCF	Inadequate control of the tumor inside PCF Brain edema due to parenchymal retraction
**Ventral Route**		
Endonasal transclival transpterygoid approach	Early tumor devascularization direct decompression of brainstem straight control of neurovascular structures	high risk of postoperative CSF leak

CSF: cerebrospinal fluid; MCF: middle cranial fossa; PCF: posterior cranial fossa; MC: Meckel’s Cave; ICA: internal carotid artery; CS: cavernous sinus.

**Table 5 cancers-14-02878-t005:** Pros and cons for each corridor for the management of cavernous sinus meningiomas (CSM).

**Corridor**	**Pros**	**Cons**
**Lateral Route**		
Anterior petrosectomy	Acceptable surgical freedom on MCF, MC, upper third of the clivus and ventral brainstem	Inadequate bony window for accessing PCF
Posterior petrosectomy	Broad view of the middle clivus Pregasserian and retrogasserian corridors Visualization of CNs from brainstem to CS Extended view over the posterior fossa	The greater the exposure the greater the risk of hearing loss and facial damage Wide exposure and manipulation of the sigmoid sinus Inadequate control of tumor extending into the MCF
Combined transpetrosal	Exploited for lesions spanning both middle and posterior cranial fossae	-
**Anterolateral Route**		
Extended pterional + extradural anterior clinoidectomy	Early unroofing of the optic nerve Early CSF release Extradural exposure of the CS	Inadequate control of tumor extending into the PCF
Fronto-temporo-orbito-zygomatic approach + extradural anterior clinoidectomy	High surgical freedom Larger exposure of lesions involving the middle fossa and adjacent areas Less brain retraction	Inadequate control of tumor extending into the PCF
Pretemporal trans-cavernous trans-Meckel’s trans-tentorial trans-petrosal	Satisfactory control of the PCF and precavernous course of CNs Good control of both supra and infra tentorial lesion	-
**Ventral Route**		
Extended endoscopic endonasal transsphenoidal Endonasal transethmoidal/transsphenoidal (far lateral) Contralateral endoscopic endonasal transsphenoidal approach	Direct visualization over the bony prominences of neurovascular structures Lesions can be followed via the thin medial wall of the CS Early pituitary gland decompression Direct infero-medial decompression of the optic canal Low risk of CSF leak	Low GTR rate

CSF: cerebrospinal fluid; MCF: middle cranial fossa; PCF: posterior cranial fossa; MC: Meckel’s Cave; ICA: internal carotid artery; CS: cavernous sinus; CNs: cranial nerves.

## Data Availability

Data is contained within the article.

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
