# Peer review of "Midline Skull Base Meningiomas: Transcranial and Endonasal Perspectives"

_cancers, 2022, doi:10.3390/cancers14122878_

Round 1
Reviewer 1 Report
The proposed article represents an extensive review of different surgical approaches for midline skull base tumors. The article is well written and based on a very comprehensive literature search. However, a few comments:
1) Recommend restructuring the article in order to provide a better overview. The results section is very short and does not provide results on the literature search except for the fact the authors found 328 articles. For example, I recommend adding to this section how many articles were found per meningeoma type and per approach. This would allow to get an idea about how well the different approaches have been studied (could be supported by a table). Also, what have been the results/outcomes of these articles.
2) The discussion section is by far too long. Again, I am wondering how the article can be re-structured in order to condense the discussion part or to provide a better overview/subsections. The research question of this review is to present a comprehensive review of indications and techniques for the management of skull base meningeomas. Maybe the headings have to be adjusted for this type of article. The discussion section should not describe the approaches per se, but include derived facts and interpretations of the different studies/outcome results found. For example, as greatly provided, the observed pros and cons of each approach. But also, if there was any paradigm change over the last decades with respect to the choice of approaches (which was well done) or whether there are any criteria related to the surgical setting, which influence the choice of approach (if observed in the analysed literature). Furthermore, the discussion should include limitations of this review - are there still knwoledge gaps in the literature and if so, which ones. Will the review influence our future practice and if so, what would be the anticipated changes or important points to consider?
3) Some tables or illustrative figures would be of advantage for summarizing some of the extensive information in a visually appealing manner. For example: figure illustrating the number of articles per meningeoma location, further splitted up by approach or a summary table indicating the tumor type, possible approaches, advantages and pitfalls or outcomes as per cited articles.
4) Paragraph numbering in section 4.4 (Cavernous sinus meningeomas) does not align with numbering of previous sections.
5) Last minor comment, list of abbreviations is missing. In the introduction you use abbreviations such as EOR and GTR. For formal correctness you mention the terms in full length first before using the abbreviation only.
Reviewer 2 Report
This is an extensive review of skull base meningiomas, focusing on the choice of surgical approaches.
- The manuscript is full of short paragraphs, which need professional English editing again.
- Abbreviations should be defined upon the first appearance (EOR, GTR...).
- The manuscript is too lengthy and describes too much basic knowledge which most neurosurgeons are familiar with.
- Suggest the authors to provide more data from the literatures, rather than description, including rate of CSF leakage, rate of complications, percentage of resection.
- Another importance parts of the skull base meningioma treatment are the WHO grades of the tumors and the subsequent treatment such as radiotherapy. These should be included. Otherwise, maximal safe resection may not lead to the best long-term outcome.
Round 2
Reviewer 1 Report
Well revised manuscript. I do not have any further hesitations with respect to its publication.
Reviewer 2 Report
The authors have revised the manuscript according to the previous comments.